# Learning Temporal 3D Semantic Scene Completion via Optical Flow Guidance

**Meng Wang**[1*] **Fan Wu**[1*] **Ruihui Li**[1†] **Yunchuan Qin**[1] **Zhuo Tang**[1†] **Kenli Li**[2,1]

[1]College of Computer Science and Electronic Engineering, Hunan University
[2]State Key Laboratory of Advanced Design and Manufacturing Technology for Vehicle, Hunan University

`{willem, wufan, liruihui, qinyunchuan, ztang, lkl}@hnu.edu.cn`
Project Page: https://github.com/willemeng/FlowScene

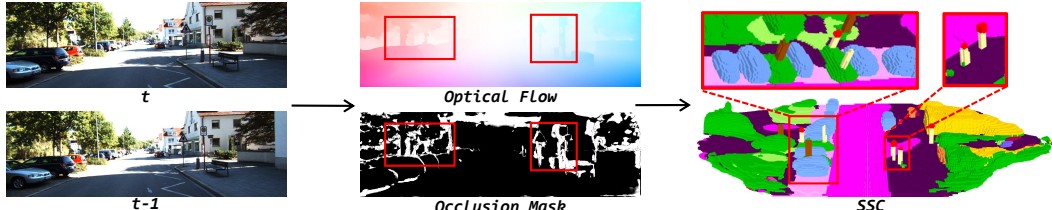

Figure 1: Given the temporal RGB images as input, our method can perform temporal modeling based on the corresponding optical flow and occlusion mask, and predict semantic scene completion for all voxels in 3D space.

## Abstract

3D Semantic Scene Completion (SSC) provides comprehensive scene geometry and semantics for autonomous driving perception, which is crucial for enabling accurate and reliable decision-making. However, existing SSC methods are limited to capturing sparse information from the current frame or naively stacking multi-frame temporal features, thereby failing to acquire effective scene context. These approaches ignore critical motion dynamics and struggle to achieve temporal consistency. To address the above challenges, we propose a novel temporal SSC method FlowScene: Learning Temporal 3D Semantic Scene Completion via Optical Flow Guidance. By leveraging optical flow, FlowScene can integrate motion, different viewpoints, occlusions, and other contextual cues, thereby significantly improving the accuracy of 3D scene completion. Specifically, our framework introduces two key components: (1) a Flow-Guided Temporal Aggregation module that aligns and aggregates temporal features using optical flow, capturing motion-aware context and deformable structures; and (2) an Occlusion-Guided Voxel Refinement module that injects occlusion masks and temporally aggregated features into 3D voxel space, adaptively refining voxel representations for explicit geometric modeling. Experimental results demonstrate that FlowScene achieves state-of-the-art performance, with mIoU of 17.70 and 20.81 on the SemanticKITTI and SSCBench-KITTI-360 benchmarks.

## 1 Introduction

One of the key challenges in autonomous driving is 3D scene understanding, which involves interpreting the spatial layout and semantic properties of objects within the scene. The ability to perceive and accurately interpret 3D scenes is essential for making safe and informed driving decisions. Recently, the 3D Semantic Scene Completion (SSC) task [29, 27] has gained significant attention in autonomous driving, as it enables the joint inference of geometry and semantics from incomplete observations.

---

*Equal contributed. †Corresponding authors.

39th Conference on Neural Information Processing Systems (NeurIPS 2025).

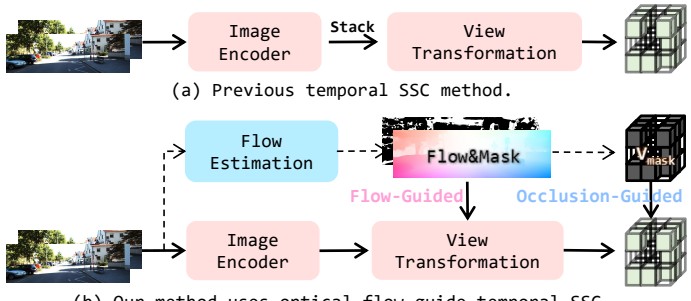

Figure 2: Our method uses optical flow guide temporal SSC versus the previous temporal SSC method.

Most existing SSC methods [26, 46, 6, 15, 27, 42] rely on input RGB images along with corresponding 3D data to predict volume occupancy and assign semantic labels. However, the dependence on 3D data often requires specialized and costly depth sensors, which can limit the broader applicability of SSC algorithms. Recently, many researchers [2, 47, 14, 9] have investigated camera-based approaches to reconstruct dense 3D geometric structures and recover semantic information, offering a more accessible alternative.

Previous camera-based SSC methods [12, 44, 14] typically rely on the limited observations available in the current frame to recover 3D geometry and semantics. Later, some researchers [17, 13, 22, 38] stacked historical temporal features or aligned features with estimated camera poses to enrich contextual information, as shown in Figure 2(a). However, these direct temporal modeling methods overlook the scene motion context, fail to achieve temporal consistency, and inherently limit the increase of effective contextual cues. Based on these limitations, we asked: ***How can we accurately identify the correlation between historical frames and the current frame to guide temporal SSC modeling?***

In this paper, we propose a novel temporal SSC method: FlowScene, Learning Temporal 3D Semantic Scene Completion via Optical Flow Guidance. As shown in Figure 2(b), FlowScene uses optical flow to guide temporal modeling, injecting various types of information into the SSC model, such as motion, different viewpoints, deformation, texture, geometric structure, lighting, and occlusion. As shown in Figure 1, the corresponding optical flow and occlusion masks are generated from the historical and current frame images, allowing for the further derivation of scene geometry and semantic structure. The positions and semantics of the car, tree trunk, vegetation, and pole within the red box in Figure 1 are more accurate, even when they are mutually occluded. Specifically, we introduce the *Flow-Guided Temporal Aggregation* module to effectively enhance temporal and motion cues by incorporating motion and contextual information from previous frames. Furthermore, we design the *Occlusion-Guided Voxel Refinement* module, which leverages aggregated features and occlusion masks to refine 3D voxel predictions for explicit geometric modeling. To evaluate the performance of FlowScene, we conduct thorough experiments on SemanticKITTI [1] and SSCBench-KITTI360 [19, 16]. Our method achieves state-of-the-art performance. The main contributions of our work are summarized as follows:

- We introduce FlowScene, a novel approach to 3D SSC that incorporates optical flow guidance to capture and model temporal and spatial dependencies across frames.

- We propose the flow-guided temporal aggregation module, which effectively enhances temporal and motion cues by incorporating motion and contextual information from previous frames.

- We design the occlusion-guided voxel refinement module, which leverages aggregated features and occlusion masks to refine 3D voxel predictions, enabling explicit geometric modeling and improving the accuracy of scene reconstruction in occluded regions.

- We evaluate FlowScene on the SemanticKITTI and SSCBench-KITTI-360 benchmarks, achieving state-of-the-art performance. Our method surpasses the latest methods in both semantic and geometric analysis, demonstrating the effectiveness of optical flow-guided temporal modeling in SSC tasks.

## 2 Related Work

**3D Semantic Scene Completion.** The vision-based 3D Semantic Scene Completion (SSC) solution has received widespread attention in the field of autonomous driving perception. MonoScene [2] was the first to infer dense 3D semantics from a single RGB image. TPVFormer [9] introduced a tri-perspective view (TPV) representation, extending BEV with two vertical planes. OccFormer [47] proposed a dual-path transformer to encode voxel features, while VoxFormer [17] introduced a two-stage pipeline for voxelized semantic scene understanding. SurroundOcc [40] employed 3D convolutions for progressive voxel upsampling and dense SSC ground truth generation. OctOcc [24] utilized an octree-based representation for semantic occupancy prediction, while NDCScene [43] re-defined spatial encoding by mapping 2D feature maps to normalized device coordinates (NDC) rather than world space. MonoOcc [48] enhanced 3D volumetric representations using an image-conditioned cross-attention mechanism. H2GFormer [39] introduced a progressive feature reconstruction strategy to propagate 2D information across multiple viewpoints. Symphonize [12] extracted high-level instance features to serve as key-value pairs for cross-attention. HASSC [38] proposed a self-distillation framework to improve the performance of VoxFormer. Stereo-based methods, such as BRGScene [14], leveraged stereo depth estimation to resolve geometric ambiguities. MixSSC [36] fused forward projection sparsity with the denseness of depth-prior backward projection. CGFormer [44] utilized a context-aware query generator to initialize context-dependent queries tailored to individual input images, effectively capturing their unique characteristics and aggregating information within the region of interest. HTCL [13] decomposed temporal context learning into two hierarchical steps: cross-frame affinity measurement and affinity-based dynamic refinement. VLScene [37] leveraged vision-language models to extract high-level semantic priors for SSC.

**Optical Flow for Visual Perception.** Optical flow estimation, a fundamental task in computer vision, aims to establish dense pixel-wise correspondences between consecutive frames. FlowNet [3, 11] introduced the first CNN-based end-to-end flow estimation pipeline, leveraging a hierarchical pyramid structure. PWC-Net [31] further refined this approach by incorporating multi-stage warping to handle large-displacement motion. RAFT [34] introduced an iterative, recurrent architecture that refines residual flow predictions in a fully convolutional manner. GMFlow [41] reframed optical flow as a global matching problem, directly computing feature similarities to establish correspondences. Beyond motion estimation, optical flow has been leveraged to enhance various vision tasks. FlowTrack [50] used optical flow to enrich feature representations and improve tracking accuracy. FGFA [49] employed flow-guided feature aggregation for end-to-end video object detection. LoSh [45] utilized flow-based warping to propagate annotations across temporal neighbors, thereby boosting referring video object segmentation. DATMO [30] introduced a moving object detection and tracking framework tailored for autonomous vehicles. DeVOS [4] incorporated optical flow into scene motion modeling, using it as a prior for learnable offsets in video segmentation.

## 3 Methodology

### 3.1 Preliminary

**Problem Setting.** Given a set of RGB images $I = \{I_{t-i}\}_{i=0}^n$, where $n$ is the number of historical temporal images, the objective is to jointly infer the geometry and semantics of a 3D scene. This scene is represented as a voxel grid $\mathbb{Y} \in \mathbb{R}^{X \times Y \times Z \times (M+1)}$, where $X, Y, Z$ represent the height, width, and depth in 3D space, respectively. Each voxel in the grid is assigned a unique semantic label from the set $C \in \{C_0, C_1, ..., C_M\}$, where $C_0$ represents empty space and the remaining classes $\{C_1, ..., C_M\}$ correspond to specific semantic categories. Here, $M$ denotes the total number of semantic classes. The goal is to learn a transformation $\mathbb{Y} = \theta(I_s)$ that closely approximates the ground truth $\hat{\mathbb{Y}}$.

### 3.2 Overview

We illustrate our method in Figure 3. First, we use the lightweight image encoder RepViT [35] and FPN [20] to extract the current image features, $F_t$, and the historical temporal features, $F_{temp} = \{F_{t-i}\}_{i=1}^n$. We then apply the pre-trained optical flow estimation model [41] (Section 3.3) to generate bidirectional optical flow, $Flow = \{Flow_{fwd}^{t-i \to t}, Flow_{bwd}^{t-i \to t}\}_{i=1}^n$. The historical temporal features, $F_{temp}$, are warped using $Flow_{bwd}$ to obtain $F_{warp} = \{F_{warp}^{t-i \to t}\}_{i=1}^n$. The bidirectional optical

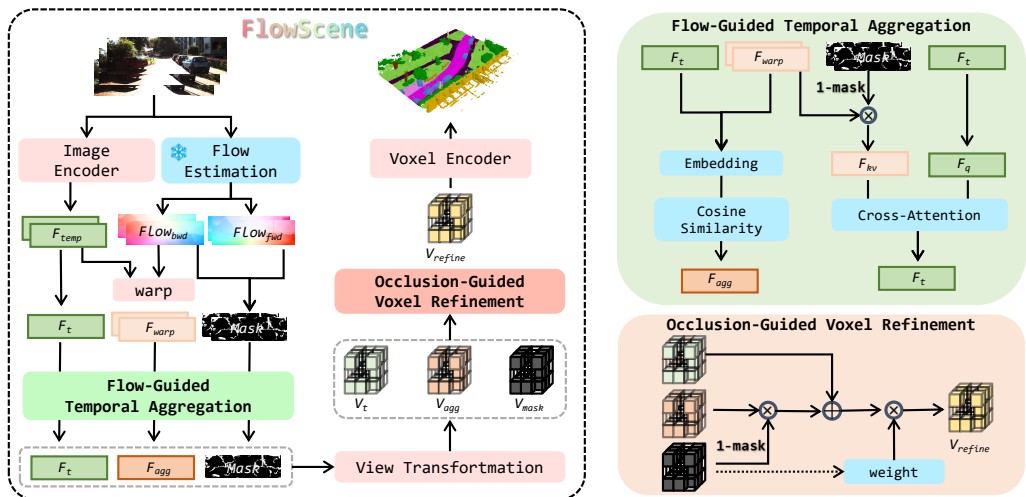

Figure 3: The FlowScene framework is proposed for temproal 3D semantic scene completion.

flow is then used for occlusion detection through a forward and backward consistency check to obtain the cumulative mask, $M \in {0,1}^{1 \times h \times w}$. Subsequently, $F_{warp}$, $F_t$, and $M$ are passed into the FGTA module (Section 3.4) to perform optical flow-guided temporal feature aggregation in the 2D image feature space, resulting in the aggregated feature $F_{agg}$. Next, we apply the LSS view transformation [25] to project $F_t$, $F_{agg}$, and $M$ into the 3D voxel space, obtaining $V_t$, $V_{agg}$, and $V_{mask}$, respectively. In the subsequent OGVR module (Section 3.5), the two voxel features are fused based on the occlusion information, yielding the refined voxel features, $V_{fine}$. Finally, $V_{fine}$ passes through the voxel encoder, then undergoes upsampling and linear projection to output the dense semantic voxels, $\mathbb{Y}$.

## 3.3 Optical Flow Estimation

**Flow-Guided Warping.** Given a reference image frame $I_t$ and historical frames $I_{t-i}$, the flow field $Flow^{t \to t-i} = \mathcal{F}(I_t, I_{t-i})$ is estimated by a flow network $\mathcal{F}$ (e.g., GMFlow [41]). The feature map from the historical frame is warped to the reference frame according to the flow. The warping function is defined as

$$F_{warp}^{t-i \to t} = \mathcal{W}arp(F_{t-i}, Flow^{t \to t-i}) \tag{1}$$

where $\mathcal{W}arp(\cdot)$ is a bilinear warping function applied to all locations of each channel in the feature map, and $F_{warp}^{t-i \to t}$ represents the feature map warped from frame $t - i$ to $t$.

**Occlusion Detection.** First, we note that there is relative motion between almost all frames in an autonomous driving scenario, which results in pixels in the current image that do not have corresponding matching pixels in the historical frames; these are referred to as occluded areas. To detect occlusion, as shown in Figure 4, we use the commonly employed forward and backward consistency check technique [32, 23], which is implemented as:

$$M = \mathcal{CC}(Flow^{t \to t-i}, Flow^{t-i \to t}), \tag{2}$$

where $\mathcal{CC}(\cdot)$ denotes the forward and backward consistency check function, and we have included a detailed explanation in the Technical Appendix. For non-occluded pixels, the forward optical flow should be the inverse of the backward optical flow of the corresponding pixel in the second frame. A pixel is marked as occluded if the mismatch between the two flows exceeds a predefined threshold. Thus, we define the occlusion flag as 1 whenever the constraint is violated and 0 otherwise.

## 3.4 Flow-Guided Temporal Aggregation

Previous SSC methods either stacked historical frame features or estimated camera poses to align features, aiming to complement the current frame. However, this direct temporal modeling approach overlooks the scene motion context, fails to achieve temporal consistency, and inherently limits the ability to leverage additional effective cues. To better incorporate time- and motion-related cues, we

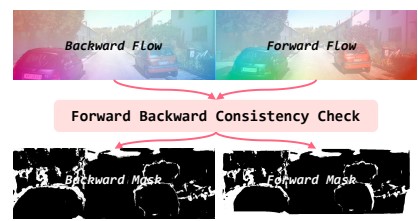

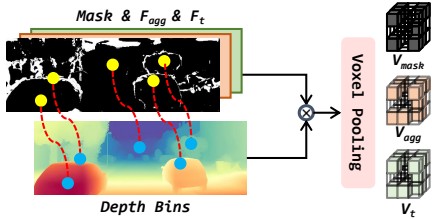

Figure 4: Occlusion detection is performed using forward-backward consistency detection.

Figure 5: Projecting the occlusion mask into 3D voxel space with depth bins.

propose a flow-guided temporal aggregation module in 2D space. This module leverages optical flow information to align and aggregate temporal features along the motion path. As illustrated on the right side of Figure 3.

Specifically, guided by optical flow, the historical frame features are warped to the reference frame. Features from different frames provide multiple information for the same object instance, such as motion, different viewpoints, deformations, textures, geometric structures, various lighting and occlusions. First, we assign different weights to different spatial locations, while ensuring that the spatial weights remain the same across all feature channels. At position $\mathbf{P}$, if the warped feature $F_{warp}^{t-i\rightarrow t}(\mathbf{P})$ is close to the feature $F_t(\mathbf{P})$, it is assigned a larger weight. Otherwise, a smaller weight is assigned. Inspired by FGFA [49], we use the cosine similarity [21] to measure the similarity between the warped features and the reference frame features:

$$w_{t-i\rightarrow t}(\mathbf{P}) = \text{similarity}(F_{warp}^{t-i\rightarrow t}(\mathbf{P}), F_t(\mathbf{P})). \tag{3}$$

Then, we use the similarity weights to aggregate these feature maps to enhance the scene motion context features. The aggregation feature $F_{agg}$ is obtained as:

$$F_{agg} = \sum_{i=0}^{t} w_{t-i\rightarrow t} \cdot F_{warp}^{t-i\rightarrow t}. \tag{4}$$

The non-occluded regions in the historical frames usually have richer texture and feature information, which may be missing in the current frame due to visual occlusion. To address this, we enhance the current frame features, we effectively fuse spatiotemporal information through the neighborhood cross-attention mechanism [7]. We selectively use non-occluded features from historical frames to prevent injecting unreliable or distorted information caused by occlusions or inaccurate flow. Including occluded regions can introduce noise and harm completion quality. First, we select reliable non-occluded region features in the historical frames based on the occlusion mask. The reference features $F_t$ are used as query, and the warp features $F_{warp}$ of the non-occluded regions serve as key and value. The specific operations are as follows:

$$F_t = \mathcal{NCA}(F_t, (1 - M) \cdot F_{warp}), \tag{5}$$

where $\mathcal{NCA}(\cdot)$ is the neighborhood cross attention mechanism. After these operations, $F_t$ fuses the non-occluded region information from both the current and historical frames, providing more stable and accurate features that enhance the perception of dynamic scenes and occluded regions.

### 3.5 Occlusion-Guided Voxel Refinement

After passing through the FGTA module, time- and motion-related cues are injected into the image features $F_t$ and the aggregate features $F_{agg}$. However, for the 3D voxel space, there is a lack of explicit geometric modeling. To incorporate occlusion and optical flow information into the 3D space, we introduce the occlusion-guided voxel refinement module. This module enhances the semantic completion ability of the occluded region by employing a weighted strategy of the occlusion mask. As shown in the right side of Figure 3.

Specifically, as shown in Figure 5, we follow the LSS view transformation paradigm and use depth bin $D$ assignment to project $F_t$, $F_{agg}$, and $M$ into the 3D voxel space to obtain $V_t$, $V_{agg}$, and $V_{mask}$, respectively,

$$\begin{aligned} V_t &= \text{VoxelPooling}(F_t \otimes D), \\ V_{agg} &= \text{VoxelPooling}(F_{agg} \otimes D), \\ V_{mask} &= \text{VoxelPooling}(M \otimes D), \end{aligned} \tag{6}$$

where $\otimes$ is the dot product operation, VoxelPooling is the voxel pooling operation. First, we use $V_{mask}$ to distinguish occluded and non-occluded regions in the 3D voxel space. For the non-occluded region, we use the aggregate features $V_{agg}$ that fuse multiple cues. Subsequently, since the information from the corresponding position in the historical frame may be inaccurate due to occlusion, we use the voxel features from the current frame to update the occluded area to supplement the latest environmental information. Finally, by constructing a weighted matrix, we normalize the fused voxel features to ensure that there is no mutation at the boundary between the occluded and non-occluded areas, thereby improving the smoothness of the features. The specific operation is as follows:

$$V_{fine} = \frac{(1 - V_{mask}) \cdot V_{agg} + V_t}{(1 - V_{mask}) + 1}. \tag{7}$$

Finally, $V_{fine}$ enters the sparse voxel encoder for feature extraction, and then performs linear prediction to output dense semantic voxels $\mathbb{Y}$.

## 3.6 Training Loss

In the FlowScene framework, we adopt the scene-class affinity loss $\mathcal{L}_{scal}$ from MonoScene [2] to optimize precision, recall, and specificity concurrently. The scene-class affinity loss is applied to semantic and geometric predictions, in conjunction with the cross-entropy loss weighted by class frequencies. Besides, the intermediate depth distribution for view transformation is supervised by the projections of LiDAR points, with the binary cross-entropy loss $\mathcal{L}_d$ following BEVDepth [18]. The overall loss function is formulated as follows:

$$\mathcal{L} = \lambda_{sem}\mathcal{L}_{scal}^{sem} + \lambda_{geo}\mathcal{L}_{scal}^{geo} + \lambda_{ce}\mathcal{L}_{ce} + \lambda_d\mathcal{L}_d, \tag{8}$$

where several $\lambda$ are balancing coefficients.

## 4 Experiments

To assess the effectiveness of our FlowScene, we conducted thorough experiments using the large outdoor datasets SemanticKITTI [1, 5], SSCBench-KITTI-360 [16, 19]. Information about datasets, metrics, and detailed implementation details is provided in the Technical Appendix, where additional experiments and analysis are also provided.

Table 1: Quantitative results on the SemanticKITTI hidden test set. The best and the second best results are in **bold** and underlined, respectively. The "S" and "T" denote single-frame images, and temporal images, respectively.

| Methods | Venues | Input | IoU | road | sidewalk | parking | other-grnd | building | car | truck | bicycle | motocycle | other-vehicle | vegetation | trunk | terrain | person | bicylist | motorcyclist | fence | pole | traf.-sign | mIoU |
|---|---|---|---|---|---|---|---|---|---|---|---|---|---|---|---|---|---|---|---|---|---|---|---|
| MonoScene [2] | CVPR'2022 | S | 34.16 | 54.70 | 27.10 | 24.80 | 5.70 | 14.40 | 18.80 | 3.30 | 0.50 | 0.70 | 4.40 | 14.90 | 2.40 | 19.50 | 1.00 | 1.40 | 0.40 | 11.10 | 3.30 | 2.10 | 11.08 |
| TPVFormer [9] | CVPR'2023 | S | 34.25 | 55.10 | 27.20 | 27.40 | 6.50 | 14.80 | 19.20 | 3.70 | 1.00 | 0.50 | 2.30 | 13.90 | 2.60 | 20.40 | 1.10 | 2.40 | 0.30 | 11.00 | 2.90 | 1.50 | 11.26 |
| OccFormer [47] | ICCV'2023 | S | 34.53 | 55.90 | 30.30 | 31.50 | 6.50 | 15.70 | 21.60 | 1.20 | 1.50 | 1.70 | 3.20 | 16.80 | 3.90 | 21.30 | 2.20 | 1.10 | 0.20 | 11.90 | 3.80 | 3.70 | 12.32 |
| Symphonize [12] | CVPR'2024 | S | 42.19 | 58.40 | 29.30 | 26.90 | 11.70 | 24.70 | 23.60 | 3.20 | 3.60 | 2.60 | 5.60 | 24.20 | 10.00 | 23.10 | 3.20 | 1.90 | 2.00 | 16.10 | 7.70 | 8.00 | 15.04 |
| BRGScene [14] | IJCAI'2024 | S | 43.34 | 61.90 | 31.20 | 30.70 | 10.70 | 24.20 | 22.80 | 2.80 | 3.40 | 2.40 | 6.10 | 23.80 | 8.40 | 27.00 | 2.90 | 2.20 | 0.50 | 16.50 | 7.00 | 7.20 | 15.36 |
| CGFormer [44] | NIPS'2024 | S | 44.41 | 64.30 | 34.20 | 34.10 | 12.10 | 25.80 | 26.10 | 4.30 | 3.70 | 1.30 | 2.70 | 24.50 | 11.20 | 29.30 | 1.70 | 3.60 | 0.40 | 18.70 | 8.70 | 9.30 | 16.63 |
| VoxFormer-T [17] | CVPR'2023 | T | 43.21 | 54.10 | 26.90 | 25.10 | 7.30 | 23.50 | 21.70 | 3.60 | 1.90 | 1.60 | 4.10 | 24.40 | 8.10 | 24.20 | 1.60 | 1.10 | 0.00 | 13.10 | 6.60 | 5.70 | 13.41 |
| H2GFormer-T [39] | AAAI'2024 | T | 43.52 | 57.90 | 30.40 | 30.00 | 6.90 | 24.00 | 23.70 | 5.20 | 0.60 | 1.20 | 5.00 | 25.20 | 10.70 | 25.80 | 1.10 | 0.10 | 0.00 | 14.60 | 7.50 | 9.30 | 14.60 |
| HASSC-T [38] | CVPR'2024 | T | 42.87 | 55.30 | 29.60 | 25.90 | 11.30 | 23.10 | 23.00 | 2.90 | 1.90 | 1.50 | 4.90 | 24.80 | 9.80 | 26.50 | 1.40 | 3.00 | 0.00 | 14.30 | 7.00 | 7.10 | 14.38 |
| SGN [22] | TIP'2024 | T | 43.71 | 57.90 | 29.70 | 25.60 | 5.50 | 27.00 | 25.00 | 1.50 | 0.90 | 0.70 | 3.60 | 26.90 | 12.00 | 26.40 | 0.60 | 0.30 | 0.00 | 14.70 | 9.00 | 6.40 | 14.39 |
| HTCL [13] | ECCV'2024 | T | 44.23 | 64.40 | 34.80 | 33.80 | 12.40 | 25.90 | 27.30 | 5.70 | 1.80 | 2.20 | 5.40 | 25.30 | 10.80 | 31.20 | 1.10 | 3.10 | 0.90 | 21.10 | 9.00 | 8.30 | 17.09 |
| **Ours** | | T | **45.20** | 64.10 | **35.00** | 33.70 | **13.00** | 27.70 | 26.40 | **10.00** | 4.20 | 3.10 | 7.00 | 26.30 | 10.00 | 30.20 | 3.10 | 5.10 | 1.10 | 20.20 | 8.90 | 9.10 | **17.70** |

## 4.1 Main Results

**Quantitative Results.** As shown in Table 1, we compare FlowScene with the latest public methods on the SemanticKITTI dataset, including approaches that use single-image input (S) and temporal image input (T). Temporal methods, such as VoxFormer [17], H2GFormer [39], HASSC [38], and SGN [22], utilize additional historical 5-frame input, while HTCL [13] uses a 3-frame historical input. In contrast, FlowScene uses only 2 historical frames as input, achieving the highest mIoU for the overall semantic metric and the highest IoU for the completion metric. Compared to the best-performing HTCL with temporal input, FlowScene improves the mIoU and IoU by 0.61%

Table 2: Quantitative results on the SSCBench-KITTI360 test set. The best and the second best results are in **bold** and underlined, respectively.

| Methods | Prec. | Rec. | IoU | car | bicycle | motocycle | truck | other-vehicle | person | road | parking | sidewalk | other-grnd | building | fence | vegetation | terrain | pole | traf.-sign | other-struct. | other-obj. | mIoU |
|---|---|---|---|---|---|---|---|---|---|---|---|---|---|---|---|---|---|---|---|---|---|---|
| MonoScene | 56.73 | 53.26 | 37.87 | 19.34 | 0.43 | 0.58 | 8.02 | 2.03 | 0.86 | 48.35 | 11.38 | 28.13 | 3.32 | 32.89 | 3.53 | 26.15 | 16.75 | 6.92 | 5.67 | 4.20 | 3.09 | 12.31 |
| VoxFormer | 58.52 | 53.44 | 38.76 | 17.84 | 1.16 | 0.89 | 4.56 | 2.06 | 1.63 | 47.01 | 9.67 | 27.21 | 2.89 | 31.18 | 4.97 | 28.99 | 14.69 | 6.51 | 6.92 | 3.79 | 2.43 | 11.91 |
| TPVFormer | 59.32 | 55.54 | 40.22 | 21.56 | 1.09 | 1.37 | 8.06 | 2.57 | 2.38 | 52.99 | 11.99 | 31.07 | 3.78 | 34.83 | 4.80 | 30.08 | 17.52 | 7.46 | 5.86 | 5.48 | 2.70 | 13.64 |
| OccFormer | 59.70 | 55.31 | 40.27 | 22.58 | 0.66 | 0.26 | 9.89 | 3.82 | 2.77 | 54.30 | 13.44 | 31.53 | 3.55 | 36.42 | 4.80 | 31.00 | 19.51 | 7.77 | 8.51 | 6.95 | 4.60 | 13.81 |
| IAMSSC | - | - | 41.80 | 18.53 | 2.45 | 1.76 | 5.12 | 3.92 | 3.09 | 47.55 | 10.56 | 28.35 | 4.12 | 31.53 | 6.28 | 29.17 | 15.24 | 8.29 | 7.01 | 6.35 | 4.10 | 12.97 |
| Symphonies | 69.24 | 54.88 | 44.12 | 30.02 | 1.85 | **5.90** | 25.07 | 12.06 | 8.20 | 54.94 | 13.83 | 32.76 | 6.93 | 35.11 | 8.58 | 38.33 | 11.52 | 14.01 | 9.57 | 14.44 | 11.28 | 18.58 |
| CGFormer | - | - | **48.07** | 29.85 | **3.42** | 3.96 | 17.59 | 6.79 | 6.63 | **63.85** | **17.15** | **40.72** | 5.53 | **42.73** | 8.22 | 38.80 | **24.94** | 16.24 | **17.45** | 10.18 | 6.77 | 20.05 |
| **Ours** | **69.70** | **58.99** | 46.95 | **32.48** | 1.87 | 4.93 | **25.47** | **14.86** | **9.62** | 60.53 | 16.49 | 36.13 | **8.58** | 39.66 | **9.62** | **39.82** | 13.32 | **17.52** | 14.35 | **16.25** | **13.08** | **20.81** |

and 0.97%, respectively. When compared to the best CGFormer, which uses single-frame input, FlowScene achieves improvements of 1.07% in mIoU and 0.79% in IoU. Additionally, our method achieves the best or second-best results in most categories, outperforming or closely matching other methods. These results demonstrate the superiority of FlowScene in both geometry and semantics, effectively utilizing optical flow motion information and achieving temporal consistency.

To demonstrate the diversity of our model, we conducted experiments on the SSCBench-KITTI-360 dataset, as shown in Table 2. It is worth noting that our method has a huge advantage in the performance of potential moving objects (such as car, truck, other-vehicle, person, etc.)

Moreover, Table 3 illustrates the performance of FlowScene across three distance ranges (12.5m, 25.6m, 51.2m) on the SemanticKITTI validation set. It is evident that our approach significantly outperforms state-of-the-art methods at every tested distance. Furthermore, as shown in Table 4, we compare the inference time and number of parameters of our method with other state-of-the-art methods on the SemanticKITTI validation set. The inference time of MonoScene is optimal because of its FLoSP feature projection method. But, FlowScene achieves state-of-the-art performance with a mIoU of 18.13%, while utilizing only 52.4M parameters. Additionally, FlowScene processes the extra 2-frame temporal image input with lower inference time, further demonstrating its efficiency and superior mIoU performance.

Table 3: Comparison of different ranges on SemanticKITTI validation set.

| Methods | Venues | mIoU(%) | | |
|---|---|---|---|---|
| | | 12.8m | 25.6m | 51.2m |
| MonoScene | CVPR'2022 | 12.25 | 12.22 | 11.30 |
| VoxFormer-T | CVPR'2023 | 21.55 | 18.42 | 13.35 |
| HASSC-T | CVPR'2024 | 24.10 | 20.27 | 14.74 |
| H2GFormer-T | AAAI'2024 | 23.43 | 20.37 | 14.29 |
| BRGScene | IJCAI'2024 | 23.27 | 21.15 | 15.24 |
| SGN-T | TIP'2024 | 25.70 | 22.02 | 15.32 |
| VLScene | AAAI'2025 | 26.51 | 24.37 | 17.83 |
| **Ours** | | **27.63** | **24.65** | **18.13** |

Table 4: Comparison of inference time and number of parameters.

| Method | Input | mIoU(%) | Times(s) | Params(M) |
|---|---|---|---|---|
| MonoScene | T | 12.96 | **0.281** | 132.4 |
| OccFormer | T | 13.58 | 0.338 | 203.4 |
| VoxFormer | T | 13.35 | 0.307 | 57.9 |
| Symphonize | S | 14.89 | 0.319 | 59.3 |
| BRGScene | S | 15.43 | 0.285 | 161.4 |
| HTCL | T | 17.13 | 0.297 | 181.4 |
| **Ours** | T | **18.13** | 0.301 | **52.4** |

**Qualitative Visualizations.** To intuitively demonstrate the performance of FlowScene, Figure 6 presents qualitative results for VoxFormer-T, BRGScene, and our method on the SemanticKITTI validation set. The first column displays the input reference image and the corresponding optical flow. It is evident that optical flow is particularly sensitive to the perception of moving objects, such as cars and cyclists. Compared to BRGScene, our method more effectively captures the location and details of mutually occluded objects in the scene (e.g., the arrangement of multiple cars in the second row). In comparison to VoxFormer-T, FlowScene maintains better temporal consistency, as shown by the car parked on the roadside in the blue box in the third row. Overall, our method demonstrates superior geometric and semantic visualization.

## 4.2 Analysis of Static and Dynamic Objects

To better understand the performance of our method across different types of semantic categories, we conduct a detailed analysis by separating static and dynamic object classes in the autonomous driving datasets. For SemanticKITTI, we define dynamic objects as the classes that are likely to

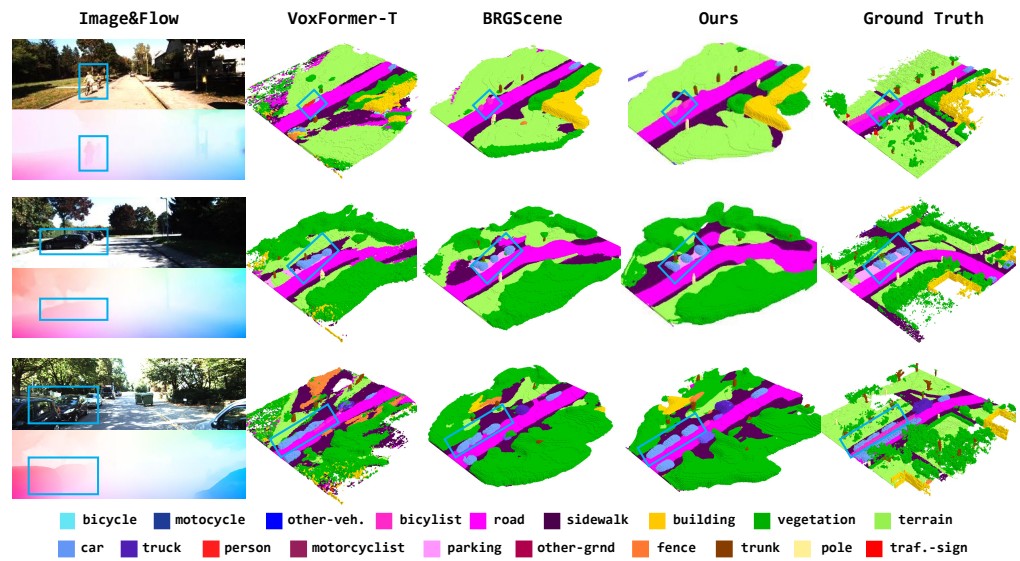

Figure 6: Qualitative results on the SemanticKITTI validation set.

Table 5: Analysis of Static and Dynamic Objects

| Method | SemanticKITTI | | | SSCBench-KITTI-360 | | |
|---|---|---|---|---|---|---|
| | Dynamic | Static | All-mIoU | Dynamic | Static | All-mIoU |
| MonoScene | 3.81 | 16.36 | 11.08 | 5.21 | 15.87 | 12.13 |
| VoxFormer | 4.45 | 19.91 | 13.41 | 4.69 | 12.19 | 11.91 |
| Symphonie | 5.71 | 21.83 | 15.04 | 13.85 | 20.94 | 18.58 |
| CGFormer | 5.48 | 24.75 | 16.63 | 11.37 | **24.38** | 20.05 |
| **Ours** | **7.50** | **25.29** | **17.70** | **14.87** | 23.78 | **20.81** |

involve motion: ⬛ car, ⬛ truck, ⬛ bicycle, ⬛ motocycle, ⬛ other-vehicle, ⬛ person and ⬛ bicylist as dynamic objects and others as static objects. For SSCBench-KITTI-360, we classified ⬛ car, ⬛ bicycle, ⬛ motocycle, ⬛ truck, ⬛ other-vehicle and ⬛ person as dynamic objects and others as static objects. Table 12 presents a comparative evaluation of our method against several prior state-of-the-art approaches, reporting mean IoU (mIoU) separately for dynamic objects, static objects, and the overall average.

Our method clearly outperforms all baselines in both datasets. Notably: On SemanticKITTI, our model achieves 7.50% mIoU for dynamic objects—the highest among all methods, surpassing CGFormer (5.48%) and Symphonie (5.71%). For static objects, we also lead with 25.29%, again outperforming CGFormer (24.75%). On SSCBench-KITTI-360, our method obtains 14.87% mIoU on dynamic objects, significantly ahead of CGFormer (11.37%) and Symphonie (13.85%). Although CGFormer slightly surpasses us on static objects (24.38% vs. 23.78%), our method achieves the highest overall score of 20.81% mIoU. These results demonstrate that FlowScene plays a crucial role in handling motion and temporal variation, making it especially effective in recognizing and completing dynamic objects. Unlike previous approaches that rely on frame stacking or static assumptions, our model adapts to motion patterns, improving semantic consistency across time.

Figure 7 illustrates the qualitative results of dynamic object modeling in a challenging real-world scenario. As shown, the cyclist in motion is effectively captured by the optical flow module, which accurately estimates the movement across frames. By leveraging this motion information, our model performs kinematic compensation, aligning temporal features and preserving object structure throughout the sequence. This results in a more coherent and complete 3D semantic reconstruction of the dynamic scene. To further support this case, we provide a supplementary video that offers an intuitive, frame-by-frame visualization of the temporal alignment and dynamic object modeling. This additional material highlights the superior capability of our method to handle complex motion compared to static-assumption baselines.

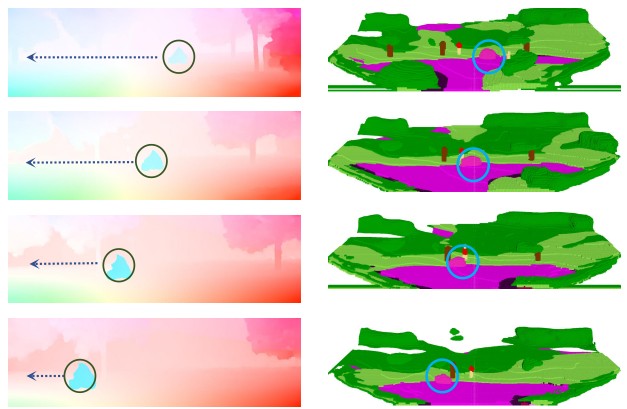

Figure 7: Dynamic object modeling case visualization results.

Table 6: Ablation study for Architecture Components on SemanticKITTI validation set. OFE: Optical Flow Estimation; FGTA: Flow-Guided Temporal Aggregation; OGVR: Occlusion-Guided Voxel Refinement; FGW: Flow-Guided Warping; OD: Occlusion Detection; TA: Temporal Aggregation; OCA: Occlusion Cross-Attention; $V_t$: reference voxel features; $V_{agg}$: aggregation voxel features; $V_{mask}$: voxel occlusion mask.

| Variants | OFE | | FGTA | | OGVR | | | IoU(%) | mIoU(%) | Params(M) |
|---|---|---|---|---|---|---|---|---|---|---|
| | FGW | OD | TA | OCA | $V_t$ | $V_{agg}$ | $V_{mask}$ | | | |
| Baseline | | | | | | | | 43.98 | 15.89 | 47.4 |
| 1 | ✓ | | | | | | | 44.13 | 16.21 | 52.1 |
| 2 | ✓ | ✓ | | ✓ | ✓ | | | 44.38 | 16.43 | 52.2 |
| 3 | ✓ | ✓ | | | ✓ | ✓ | ✓ | 44.56 | 16.67 | 52.1 |
| 4 | ✓ | ✓ | ✓ | | ✓ | ✓ | ✓ | 44.63 | 17.23 | 52.3 |
| 5 | ✓ | ✓ | | ✓ | ✓ | ✓ | ✓ | 44.42 | 17.08 | 52.3 |
| 6 | ✓ | ✓ | ✓ | ✓ | ✓ | | | 44.68 | 17.18 | 52.4 |
| 7 | ✓ | ✓ | ✓ | ✓ | ✓ | ✓ | | 44.72 | 17.63 | 52.4 |
| 8 | ✓ | ✓ | ✓ | ✓ | ✓ | ✓ | ✓ | **45.01** | **18.13** | 52.4 |

## 4.3 Ablation Studies

We conduct extensive ablation experiments for FlowScene on the Semantickitti validation set. Specifically, we analyze the impact of different architecture component variations in Table 6.

**Optical Flow Estimation (OFE).** The baseline model removes all components, using only the current image and two frames of historical images as input. After passing through the image encoder, all features are stacked together. Variant 1 in Table 6 uses Flow-Guided Warping to align the temporal features to the reference moment, achieving a 0.32% mIoU improvement (Variant 1 vs. Baseline). Additionally, Variant 2 incorporates Occlusion Detection to obtain an occlusion mask, which guides the interaction of non-occluded areas in the 2D feature space, boosting the mIoU score by 0.22% (Variant 2 vs. Variant 1).

**Flow-Guided Temporal Aggregation (FGTA).** In Table 6, Variants 3, 4, and 5 represent different configurations of the FGTA module: removing the FGTA module (Variant 3), removing the Occlusion Cross-Attention (Variant 4), and removing the Temporal Aggregation component (Variant 5). Variant 4 adaptively assigns weights to aggregate historical features, resulting in a 0.56% mIoU improvement (Variant 4 vs. Variant 3). Variant 5 uses Occlusion Cross-Attention to facilitate interaction between the current feature and the non-occluded areas in the historical frame, enhancing the texture and contextual information of the current frame's features, further boosting the mIoU by 0.41% (Variant 5 vs. Variant 3).

**Occlusion-Guided Voxel Refinement (OGVR).** In Table 6, Variant 6 represents the removal of the OGVR module, while Variant 7 uses convolution fusion to concatenate $V_t$ and $V_{agg}$. Even with this simple fusion strategy, a 0.45% mIoU improvement is achieved (Variant 7 vs. Variant 6). Variant 8 represents our final full model. Compared to Variant 7, the mask-based refinement strategy further improves the mIoU metric. It is worth noting that the OGVR module incurs no additional parameter overhead.

Table 7: Ablation study for temporal alignment strategy.

| Method | IoU(%) | mIoU(%) |
|---|---|---|
| Stack | 43.98 | 15.89 |
| VoxFormer-T [17] | 44.15 | 13.35 |
| HTCL [13] | **45.51** | 17.13 |
| Flow-Guided [Ours] | 45.01 | **18.13** |

Table 8: Ablation study for different number of temporal inputs.

| Temporal Inputs | | | | | IoU(%) | mIoU(%) | Times(s) |
|---|---|---|---|---|---|---|---|
| t-1 | t-2 | t-3 | t-4 | t-5 | | | |
| ✓ | | | | | 44.63 | 17.74 | 0.290 |
| ✓ | ✓ | | | | 45.01 | 18.13 | 0.301 |
| ✓ | ✓ | ✓ | | | 44.72 | 18.30 | 0.314 |
| ✓ | ✓ | ✓ | ✓ | | 44.66 | 17.68 | 0.328 |
| ✓ | ✓ | ✓ | ✓ | ✓ | 44.53 | 17.55 | 0.344 |

Table 9: Ablation study for optical flow networks.

| Method | IoU(%) | mIoU(%) | Params(M) |
|---|---|---|---|
| PWC-Net+ [31] | 43.31 | 17.13 | 8.8 |
| RAFT [34] | 44.12 | 17.56 | 5.3 |
| FlowFormer [10] | 44.33 | 17.74 | 18.2 |
| GMFlow [41] | **45.01** | **18.13** | **4.7** |

Table 10: Ablation study for image backbone networks. Acc represents the classification accuracy of each pre-trained model on ImageNet [28].

| Method | IoU(%) | mIoU(%) | Params(M) | Acc(%) |
|---|---|---|---|---|
| ResNet50 [8] | 44.12 | 16.98 | 25.6 | 79.3 |
| EfficientNetB7 [33] | 44.31 | 17.63 | 63.8 | **84.4** |
| RepVit-M2.3 [35] | **45.01** | **18.13** | **22.4** | 83.3 |

Overall, compared to the baseline, our method achieves significant improvements in both completion and semantic metrics (+2.24% mIoU, +1.03% IoU).

**Temporal Alignment Strategy and Temporal Inputs.** Table 7 presents the results of an ablation study comparing different temporal alignment strategies used for fusing features across frames. Our Flow-Guided strategy delivers the best mIoU of 18.13%, showing that optical flow-guided alignment is highly effective for preserving fine-grained semantic consistency, particularly for dynamic scenes. As shown in Table 8, we evaluate the performance of temporal inputs with different numbers of frames. We observe that, as the number of frames increases, the time overhead also increases. However, the mIoU metric does not grow linearly, as the quality of optical flow prediction decreases when the time interval between frames is longer. As a result, inputs with 4 or 5 frames (t-4 and t-5) lead to reduced effectiveness. Considering both the experimental metrics and the time overhead, we use 2 frames as the input for our FlowScene method.

**Optical Flow Networks.** Table 9 presents the performance of different optical flow networks. We compare several state-of-the-art methods, including PWC-Net [31], RAFT [34], and FlowFormer [10], along with our setting, GMFlow [41], which is highlighted in the last row. Our setting achieves the highest IoU of 45.01% and mIoU of 18.13%, outperforming all other methods in both metrics. These results suggest that GMFlow effectively captures motion cues and integrates them into the semantic scene completion task, providing superior performance over the other optical flow networks tested, with significantly fewer parameters.

**Image Backbone Networks.** Table 10 examines the impact of different backbone networks on the performance of FlowScene. The study compares EfficientNetB7 [33], ResNet50 [8], and RepVit-M2.3 [35](our setting). Our method, using RepVit-M2.3, achieves the highest IoU of 45.01% and mIoU of 18.13%, surpassing both EfficientNetB7 (44.31% IoU, 17.63% mIoU) and ResNet50 (44.12% IoU, 16.98% mIoU). RepVit-M2.3, though achieving the best performance, maintains a relatively low parameter count of 22.4M. In comparison, EfficientNetB7 has a much higher parameter count of 63.8M, while ResNet50 is more parameter-efficient at 25.6M. RepVit-M2.3 offers a good balance between performance and parameter count, making it an ideal choice for our backbone network. Different image encoders have significant improvements over the baseline, demonstrating the effectiveness of our entire model.

## 5  Conclusion

In this paper, we propose a novel temporal SSC method FlowScene. Specifically, we introduce a Flow-Guided Temporal Aggregation module that aligns and aggregates temporal features using optical flow, capturing motion-aware context and deformable structures. In addition, we design an Occlusion-Guided Voxel Refinement module that injects occlusion masks and temporally aggregated features into 3D voxel space, adaptively refining voxel representations for explicit geometric modeling. Experimental results demonstrate that FlowScene achieves SOTA performance on the SemanticKITTI and SSCBench-KITTI-360 benchmarks.

## Acknowledgments and Disclosure of Funding

This work was supported in part by the State Key Laboratory of Advanced Design and Manufacturing Technology for Vehicle under Grant 72365001, in part by the National Natural Science Foundation of China(Grant Nos. 62225205, 62472162, 62473137), in part by the National Key Researchand Development Program of China (No.2025YFB3003601), in part by the Major Science and Technology Research Projects of Hunan Province (Grant Nos. 2024QK2010, 2024QK2009).

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

## Technical Appendices and Supplementary Material

This technical appendices consists of the following sections:

- In Section A, we provide information of datasets, metrics and detailed implementation details.
- In Section B, we provide more experiments results and analysis.
- In Section C, we analyze the limitations of our approach, directions for future work, and the broader impacts of this work.

We also include a video in the supplementary material.

## A  Experimental Setup

**Datasets.**    The SemanticKITTI[1, 5] dataset includes dense semantic scene completion annotations and labels a voxelized scene with 20 semantic classes. It consists of 10 training sequences, 1 validation sequence, and 11 testing sequences. RGB images are resized to $1280 \times 384$ for input processing. The SSCBench-KITTI-360[16, 19] dataset contains 7 training sequences, 1 validation sequence, and 1 testing sequence, covering 19 semantic classes in total. The RGB images are resized to $1408 \times 384$ for input processing.

**Metrics.**    We use intersection over union (IoU) to evaluate the scene completion performance. To assess the effectiveness of our 3D Semantic Scene Completion method, we focus on the mean IoU (mIoU). A higher IoU value reflects accurate geometric predictions, while a higher mIoU value indicates more precise semantic segmentation.

**Training Details.**    We use RepVit [35] and FPN [20] to extract features for all images. The number of historical temporal frames n is set to 2. We use and freeze the GMFlow [41] optical flow estimation model to obtain optical flow information. We use the LSS paradigm for 2D-3D projection. The neighborhood cross-attention range is set to 7, and the number of attention heads is set to 8. Finally, the final outputs of SemantiKITTI is 20 classes, and SSCBench-KITTI-360 is 19 classes. All datasets have the scene size of $51.2m \times 51.2m \times 64m$ with the voxel grid size of $256 \times 256 \times 32$. By default, the model is trained for 25 epochs. We optimise the process, utilizing the AdamW optimizer with an initial learning rate of 1e-4 and a weight decay of 0.01. We also employ a multi-step scheduler to reduce the learning rate. All models are trained on two A100 Nvidia GPUs with 80G memory and batch size 4.

**Implementation of Flow Consistency Check.**

**1. Variable Definition:**

**Forward Flow**: ( $Flow^{t \to t-1}$ )

- Maps pixels from the current frame ( $I_t$) to the previous frame ( $I_{t-1}$ ).
- For a pixel ($x_t \in I_t$), the corresponding location in ($I\_t - 1$) is:
  $x_{t-1} = x_t + Flow^{t \to t-1}(x_t)$

**Backward Flow**: ($Flow^{t-1 \to t}$ )

- Maps pixels from the previous frame ($I_{t-1}$) to the current frame ( $I_t$ ).
- For a pixel ( $x_{t-1} \in I\_t - 1$), the corresponding location in ( $I_t$ ) is:
  $x'_t = x\_t - 1 + Flow^{t-1 \to t}(x\_t - 1)$

**2. Consistency Check**

The **forward-backward consistency check** verifies whether a pixel mapping is valid by ensuring round-trip correspondence.

**Round-trip Mapping**

A pixel in ( $I_t$ ) is mapped to ( $I_{t-1}$ ) using forward flow, and then mapped back using backward flow:

$x_t'' = x_t + Flow^{t \to t-1}(x_t) + Flow^{t-1 \to t}(x_t + F^{t \to t-1}(x_t))$

Define the **consistency residual** as:
$\Delta(x_t) = Flow^{t \to t-1}(x_t) + Flow^{t-1 \to t}(x_t + F^{t \to t-1}(x_t))$

If the norm ( $\|\Delta(x_t)\|$ ) is small (below a threshold), the mapping is considered consistent.

### 3. Occlusion Mask

Pixels with high inconsistency are typically considered **occluded or unreliable**.

**Occlusion Mask ($M(x)$):**

$$M(x) = \begin{cases} 1 & \text{if } \|\Delta(x)\| > \tau \quad \text{(occluded)} \\[2em] 0 & \text{otherwise} \quad \text{(non-occluded)} \end{cases}$$

Where ( $\tau$ ) is a predefined threshold.

**Implementation of FGTA and OGVR module.** Algorithms 1 and 2 describe the implementation details of the two key components proposed in this work: Flow-Guided Temporal Aggregation (FGTA) and Occlusion-Guided Voxel Refinement (OGVR).

Algorithm 1 outlines the inference procedure for FGTA. Given the current and historical image frames $\{I_{t-i}\}_{i=0}^N$ and their corresponding features $\{F_{t-i}\}_{i=0}^N$, the algorithm first estimates forward and backward optical flows between the current frame $I_t$ and each historical frame $I_{t-i}$. Each historical feature map $F_{t-i}$ is warped to the reference frame using flow-guided warping. Cosine similarity between the warped features and the current frame feature $F_t$ is computed to assign adaptive weights, which emphasize temporally consistent and visually similar regions. Simultaneously, an occlusion mask is constructed through a bidirectional flow consistency check to identify unreliable regions. The weighted historical features are aggregated into $F_{agg}$, and a Neighborhood Cross-Attention (NCA) operation is applied to further refine $F_t$, focusing only on non-occluded regions. This process enhances temporal consistency and robustness to motion and occlusion artifacts.

Algorithm 2 describes the inference procedure for the OGVR module. The goal is to refine the voxel features in 3D space by integrating information from the aggregated feature volume $V_{agg}$, the current frame's voxel features $V_t$, and the occlusion mask volume $V_{mask}$. Non-occluded regions are updated using features from $V_{agg}$, while occluded regions are filled in using the current frame's voxel features $V_t$, which are more reliable in such areas. A per-voxel weight map is maintained to track the number of valid sources contributing to each voxel. The final voxel feature representation $V_{fine}$ is obtained by normalizing the fused features with the accumulated weights, ensuring smooth transitions at the boundaries between occluded and non-occluded regions. Importantly, this refinement process is lightweight and introduces no additional parameter overhead.

Together, these modules enable FlowScene to effectively align temporal information and reason across occlusions in both 2D and 3D spaces, leading to improved semantic and geometric scene understanding in dynamic environments.

## B    More Results

### B.1    Reproduce SOTA Method using Different Image Encoder

To ensure a fair and consistent comparison across methods, we re-implemented several state-of-the-art Semantic Scene Completion models using a unified experimental setup. Specifically, we replaced the original image encoders used in existing methods with a lightweight and efficient backbone—RepViT [35]—while keeping all other architectural and training settings unchanged.

As shown in Table 11, the results demonstrate that RepViT significantly reduces the number of parameters across all models, often without degrading performance—and in some cases, even improving it. This confirms the generalizability and efficiency of RepViT as an image encoder for SSC tasks.

---

**Algorithm 1** Inference algorithm of flow-guided temporal alignment

---

1: **Inputs:** Images $\{I_{t-i}\}_{i=0}^{N}$, Features $\{F_{t-i}\}_{i=0}^{N}$
2: $M = Zeros$                                                                  ▷ init occlusion mask
3: **for** $i = 1$ to $N$ **do**
4:     $Flow^{t \to t-i}, Flow^{t-i \to t} \leftarrow \mathcal{F}(I_t, I_{t-i})$                         ▷ compute dual optical flow
5:     $F_{warp}^{t-i \to t} \leftarrow \mathcal{W}arp(F_{t-i}, Flow^{t \to t-i})$                              ▷ flow-guided warp
6:     $w_{t-i \to t} \leftarrow \text{similarity}(F_{warp}^{t-i \to t}, F_t)$                         ▷ compute similarity weight
7:     $F_{agg}[i] \leftarrow w_{t-i \to t} \cdot F_{warp}^{t-i \to t}$               ▷ get the features corresponding to the weights
8:     $M \leftarrow \mathcal{CC}(Flow^{t \to t-i}, Flow^{t-i \to t}) \cup M$                 ▷ compute occlusion mask
9: **end for**
10: $F_{agg} \leftarrow \text{SUM}(F_{agg})$
11: $F_t \leftarrow \mathcal{NCA}(F_t, (1 - M) \cdot F_{warp})$
12: **Outputs:** $F_{agg}, F_t, M$

---

---

**Algorithm 2** Inference algorithm of occlusion-guided voxel refinement

---

1: **Inputs:** Aggregated voxel feature $V_{agg}$, current voxel feature $V_t$, occlusion mask $V_{mask}$
2: Initialize weight $\leftarrow \mathbf{0}$ with shape as $V_t$
3: Initialize $V_{fine} \leftarrow \mathbf{0}$ with shape as $V_t$
                                                                                     ▷ fuse non-occluded regions
4: $V_{fine} \leftarrow V_{fine} + V_{agg} \cdot (1 - V_{mask})$
5: weight $\leftarrow$ weight $+ (1 - V_{mask})$
                                                                                     ▷ fuse occluded regions
6: $V_{fine} \leftarrow V_{fine} + V_t$
7: weight $\leftarrow$ weight $+ 1$
                                                                                     ▷ normalize result
8: weight $\leftarrow \max(\text{weight}, 1 \times 10^{-6})$
9: $V_{fine} \leftarrow V_{fine}/\text{weight}$
10: **Outputs:** Refined voxel feature $V_{fine}$

---

### B.2 Standard Errors and Standard Deviations Results

We conducted a statistical analysis on the SemanticKITTI validation set and report the **weighted mean IoU (W-mIoU)**, **weighted standard deviation (W-SD)**, and **weighted standard error (W-SE)** across semantic categories. As shown, our method not only achieves the **highest mIoU and W-mIoU**, but also demonstrates **W-SD** and **W-SE** compared to other strong baselines. This indicates that our performance improvements are statistically meaningful and stable across classes.

### B.3 More Quantitative Results

To provide a more thorough comparison, we provide additional quantitative results of semantic scene completion on the SemanticKITTI validation set in Table 13. The results further demonstrate the effectiveness of our approach in enhancing 3D scene perception performance. Compared with the previous state-of-the-art methods, FlowScene is superior to other HTCL [13] in semantic scene understanding, with a 1.00% increase in mIoU. In addition, compared with Symphonize [12], huge improvements are made in both occupancy and semantics. IoU and mIoU enhancement are of great significance for practical applications. It proves that we are not simply reducing a certain metric to achieve semantic scene completion.

### B.4 Failure Case

We provide two failure cases in Figure 8.

### B.5 More Visualizations Results

We show visualization examples on the Semantickitti validation set, as shown in Figure 9. From left to right are the input image, the corresponding optical flow and occlusion mask, the front view

Table 11: Reproduce SOTA method using different image encoder.

| Method | Backbone | mIoU(%) | Params(M) |
|---|---|---|---|
| MonoScene [2] | EfficientNetB7 | 12.96 | 132.4 |
| | RepViT | 12.59 | 91.0 |
| VoxFormer [17] | ResNet50 | 13.35 | 57.9 |
| | RepViT | 13.62 | 54.7 |
| BRGScene [14] | EfficientNetB7 | 15.43 | 161.4 |
| | RepViT | 16.13 | 120.0 |
| HTCL [13] | EfficientNetB7 | 17.13 | 181.4 |
| | RepViT | 16.86 | 140.0 |
| VLScene [37] | EfficientNetB7 | 17.44 | 88.8 |
| | RepViT | 17.83 | **47.4** |
| Ours | EfficientNetB7 | 17.63 | 93.8 |
| | ResNet50 | 16.98 | 55.6 |
| | RepViT | **18.13** | 52.4 |

Table 12: Standard Errors and Standard Deviations Results

| Method | mIoU(↑) | W-mIoU(↑) | W-SD(↓) | W-SE(↓) |
|---|---|---|---|---|
| **Ours** | **17.70** | **33.11** | **13.80** | **6.50** |
| HTCL (ECCV'2024) | 17.08 | 32.64 | 14.17 | 6.67 |
| CGFormer (NIPS'2024) | 16.63 | 31.91 | 14.43 | 6.80 |
| BRGScene (IJCAI'2024) | 15.35 | 30.22 | 14.01 | 6.60 |
| TPVFormer (ICCV'2023) | 12.32 | 24.44 | 14.34 | 6.75 |
| MonoScene (CVPR'2022) | 11.08 | 22.58 | 14.38 | 6.77 |

SSC, and the top view SSC. Due to the motion information brought by the optical flow, the location information of the scene objects is more accurate and the layout is more reasonable. We report the performance of more visual comparison results on the SemanticKITTI validation set in Figure 10. We compare with VoxFormer [17] and BRGScene [14]. In general, our method performs more fine-grained segmentation of the scene and maintains clear segmentation boundaries. For example, in the segmentation completion result of cars, we predict clear separation of each car. In contrast, other methods show continuous semantic errors for occluded cars. In addition, our flow can effectively deal with the problem of mutual occlusion between different objects. Finally, we provide a video in the appendix to show the performance more intuitively.

# C    Discussions

## C.1    Limitations

Flowscene shows strong performance on the benchmark with an improved number of parameters. This is beneficial for deploying real-world autonomous driving applications. But the inference time of the model needs to be improved. While optical flow is effective, it depends on pretrained flow model, potentially limiting performance in degraded visual conditions.

## C.2    Future Works

Semantic scene completion in multi-camera settings is also worth attention, which is our future work. Meanwhile, the legal challenges of autonomous driving as well as privacy and data security risks are still topics of debate. Finally, the robustness of semantic scene completion is also an issue worth exploring.

## C.3    Broader Impacts

FlowScene enhances 3D geometry perception by aligning temporal features through optical flow information, thereby improving the ability of temporal semantic scene completion. This work has a non-obvious negative social impact.

Table 13: Quantitative results on the SemanticKITTI validation set. The best and the second best results are in **bold** and underlined, respectively.

| Methods | Published | Inputs | IoU | road (15.30%) | sidewalk (11.13%) | parking (1.12%) | other-grnd (0.56%) | building (14.10%) | car (3.92%) | truck (0.16%) | bicycle (0.03%) | motocycle (0.03%) | other-vehicle (0.20%) | vegetation (39.3%) | trunk (0.51%) | terrain (9.17%) | person (0.07%) | bicylist (0.07%) | motorcyclist (0.05%) | fence (3.90%) | pole (0.29%) | traf.-sign (0.08%) | mIoU |
|---|---|---|---|---|---|---|---|---|---|---|---|---|---|---|---|---|---|---|---|---|---|---|---|
| MonoScene [2] | CVPR'2022 | S | 36.86 | 56.52 | 26.72 | 14.27 | 0.46 | 14.09 | 23.26 | 6.98 | 0.61 | 0.45 | 1.48 | 17.89 | 2.81 | 29.64 | 1.86 | 1.20 | 0.00 | 5.84 | 4.14 | 2.25 | 11.08 |
| TPVFormer [9] | CVPR'2023 | S | 35.61 | 56.50 | 25.87 | 20.60 | 0.85 | 13.88 | 23.81 | 8.08 | 0.36 | 0.05 | 4.35 | 16.92 | 2.26 | 30.38 | 0.51 | 0.89 | 0.00 | 5.94 | 3.14 | 1.52 | 11.36 |
| OccFormer[47] | ICCV'2023 | S | 36.50 | 58.85 | 26.88 | 19.61 | 0.31 | 14.40 | 25.09 | 25.53 | 0.81 | 1.19 | 8.52 | 19.63 | 3.93 | 32.62 | 2.78 | 2.82 | 0.00 | 5.61 | 4.26 | 2.86 | 13.46 |
| Symphonize [12] | CVPR'2024 | S | 41.92 | 56.37 | 27.58 | 15.28 | 0.95 | 21.64 | 28.68 | 20.44 | 2.54 | 2.82 | 13.89 | 25.72 | 6.60 | 30.87 | 3.52 | 2.24 | 0.00 | 8.40 | 9.57 | 5.76 | 14.89 |
| VoxFormer-T[17] | CVPR'2023 | T | 44.15 | 53.57 | 26.52 | 19.69 | 0.42 | 19.54 | 26.54 | 7.26 | 1.28 | 0.56 | 7.81 | 26.10 | 6.10 | 33.06 | 1.93 | 1.97 | 0.00 | 7.31 | 9.15 | 4.94 | 13.35 |
| H2GFormer [39] | AAAI'2024 | T | 44.69 | 57.00 | 29.37 | 21.74 | 0.34 | 20.51 | 28.21 | 6.80 | 0.95 | 0.91 | 9.32 | **27.44** | 7.80 | 36.26 | 1.15 | 0.10 | 0.00 | 7.98 | 9.88 | 5.81 | 14.29 |
| HASSC [38] | CVPR'2024 | T | 44.58 | 55.30 | 29.60 | **25.90** | **11.30** | 23.10 | 23.00 | 2.90 | 1.90 | 1.50 | 4.90 | 24.80 | **9.80** | 26.50 | 1.40 | **3.00** | 0.00 | **14.30** | 7.00 | **7.10** | 14.74 |
| HTCL [13] | ECCV'2024 | T | **45.51** | 63.70 | **32.48** | 23.27 | 0.14 | 24.13 | **34.30** | 20.72 | **3.99** | 2.80 | 11.99 | 26.96 | 8.79 | **37.73** | 2.56 | 2.70 | 0.00 | 11.22 | 11.49 | 6.95 | 17.13 |
| **Ours** | | T | 45.01 | **63.72** | 32.10 | 22.20 | 1.31 | **25.63** | 33.33 | **33.47** | 2.36 | **5.09** | **16.99** | 26.35 | 8.68 | 36.73 | **3.79** | 1.92 | 0.00 | 12.05 | **11.65** | 7.05 | **18.13** |

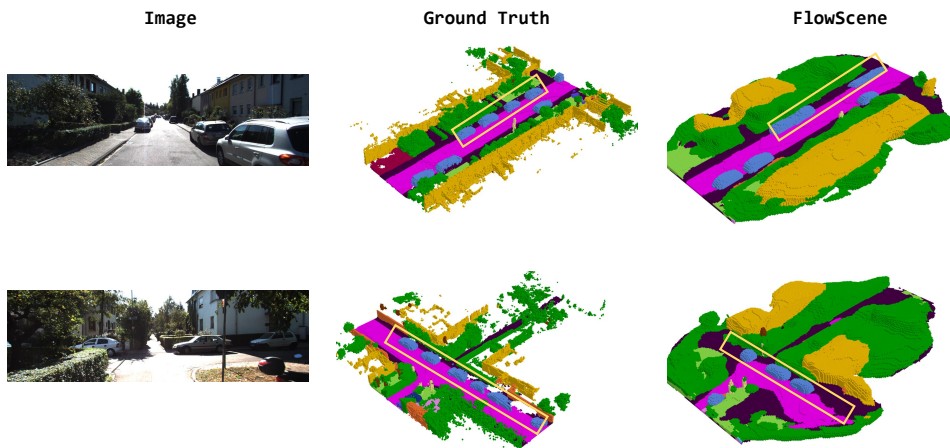

Figure 8: Failure cases.

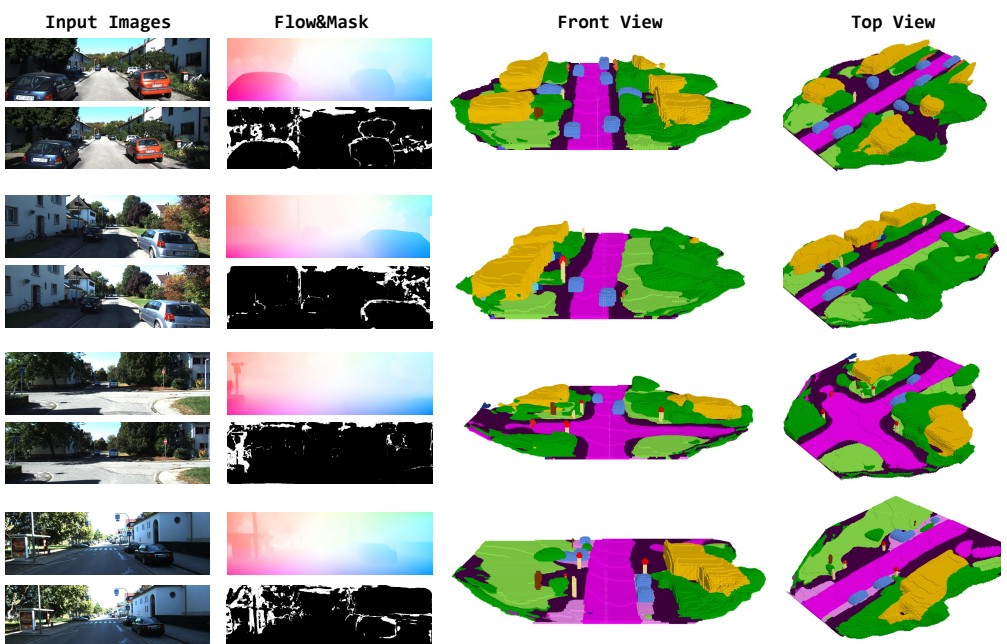

Figure 9: Qualitative results on the SemanticKITTI validation set.

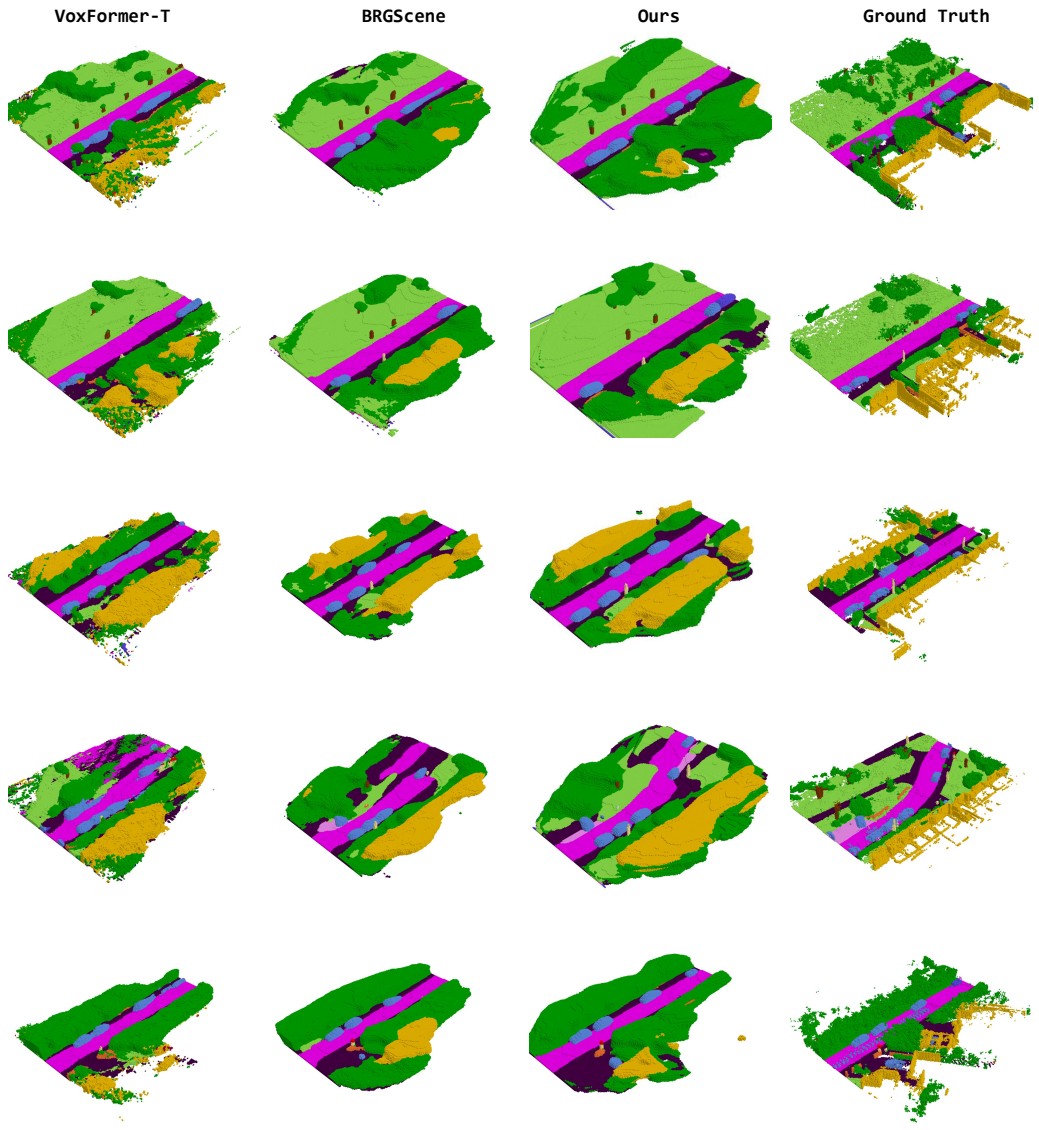

Figure 10: Qualitative results on the SemanticKITTI validation set.

