# OpenReview forum: "Learning Temporal 3D Semantic Scene Completion via Optical Flow Guidance"
_NeurIPS.cc/2025/Conference — NeurIPS 2025 poster_

### Official Review · Reviewer_n69C · 2025-06-28

**Clarity:** 3
**Significance:** 3
**Originality:** 3
**Rating:** 4
**Confidence:** 4

**Summary:**

This paper presents FlowScene, a novel framework for temporal 3D Semantic Scene Completion that explicitly uses optical flow to guide spatio-temporal dependencies. Specifically, the Flow-Guided Temporal Aggregation module is designed to align features from past frames using motion cues, while the Occlusion-Guided Voxel Refinement module intelligently combines current and historical information using occlusion masks. Experiments on SemanticKITTI and SSCBench-KITTI-360 benchmarks demonstrate the effectiveness of the proposed modules.

**Questions:**

It would be better to provide more evidence to address my concerns about the role of occlusion masks in refining voxel features and the limited improvement on the KITTI360 test set. I would raise the evaluation score when these concerns are solved.

**Ethical Concerns:**

["NO or VERY MINOR ethics concerns only"]

**Final Justification:**

The authors addressed my primary concerns. Considering the feedback from other reviewers, I am inclined to raise my score to borderline accept.

**Limitations:**

yes

**Quality:**

3

**Strengths And Weaknesses:**

**Strengths**:
- Utilizing optical flow as explicit guidance for temporal feature aggregation provides more fine-grained motion and contextual cues.
- The introduction of occlusion masks in 3D voxel refinement enhances representation quality in occluded regions.
- Extensive experimental validation with detailed ablation studies demonstrates the effectiveness of each component.

**Weaknesses**:
- The main concern is about the role of occlusion masks in refining voxel features. According to the paper's description, occlusion masks are primarily used to enhance the fusion of features from non-occluded regions in historical frames, while using current frame features for occluded parts. However, for constructing 3D voxels of the current frame, it's more reasonable and helpful to obtain 3D features about the occluded parts of the current frame from historical frames to achieve a more complete 3D representation. Conversely, when performing 2D feature fusion, it's better to consider the interference of occluded features, and for the current frame, it will make sense to fuse non-occluded 2D features.
- It would be better to provide more explanation on the limited improvement on the KITTI360 test set compared to the SOTA.
- It would be better to unify the notation of occlusion masks in Figure 3 and Figure 4 for better clarification.
- Based on the supplementary video materials, the occlusion masks generated from optical flow appear to have significant errors. It would be beneficial to further elaborate on the role and significance of generating occlusion masks through optical flow.

---

> ### Author Rebuttal · Authors · 2025-07-27
>
> *We sincerely thank the reviewer for the valuable feedback and insightful suggestions. Below, we address the identified weaknesses and questions in detail.*
>
>
>
> ------
>
> > **[Weakness 1] The main concern is about the role of occlusion masks in refining voxel features. According to the paper's description, occlusion masks are primarily used to enhance the fusion of features from non-occluded regions in historical frames, while using current frame features for occluded parts. However, for constructing 3D voxels of the current frame, it's more reasonable and helpful to obtain 3D features about the occluded parts of the current frame from historical frames to achieve a more complete 3D representation. Conversely, when performing 2D feature fusion, it's better to consider the interference of occluded features, and for the current frame, it will make sense to fuse non-occluded 2D features.**
>
> **[Answer]** *We sincerely thank the reviewer for the valuable suggestion. We explain this issue as follows:*
>
> 1. *In the **early stages of our study**, we indeed implemented a similar design that uses features from historical frames to supplement the occluded regions in the current frame. However, our experiments showed that this strategy **did not improve the performance** and, in scenes with significant object motion, even led to performance degradation.*
>
> 2. *Upon further **investigation and theoretical analysis**, we realized that autonomous driving scenarios are characterized by **forward ego-motion** and **frequent dynamic objects**. As a result, the occluded regions in the current frame are often **not visible in historical frames**, or the features from past frames may suffer from **severe geometric misalignment and semantic inconsistency** due to perspective shifts and object movement.*
>
> 3. *In our implementation, the “occlusion regions” are identified using **forward-backward flow consistency check**, which essentially marks **uncertain areas around object boundaries**. After being projected into the 3D voxel space, these regions become **unreliable regions**, and any features obtained from previous frames in these locations are equally **untrustworthy**. Therefore, we choose to fill these regions using the **current frame’s features**, which are spatially aligned and directly observed, making them more geometrically **accurate and stable**.*
>
> 4. *Moreover, consistent with observations from prior multi-frame fusion works, **features of dynamic objects or low-texture areas** are highly prone to drift and deformation across time, whereas features extracted from the current frame are generally more **robust and reliable** in such regions.*
>
> 5. *For 2D feature fusion, we use the **cosine similarity** between the current frame feature and the warped historical frame feature to design the weight. When the warp historical frame feature $F_{warp}$ is close to the current feature $F_t$, a larger weight is assigned, otherwise a smaller weight (i.e. when **the optical flow error is large**) is assigned **(see Sec. 3.4).** Then, we employ a **occlusion cross-attention** to effectively integrate the features in **non-occluded regions**.*
>
>
>
> ------
>
> > **[Weakness 2] It would be better to provide more explanation on the limited improvement on the KITTI360 test set compared to the SOTA.**
>
> ***[Answer]** Thank you for this important observation. While our method achieves slightly lower IoU compared to CGFormer, we would like to highlight several key distinctions that explain this and emphasize our method’s strengths:*
>
> 1. ***Use of Pretrained Depth in CGFormer:***
>    *CGFormer benefits from **the pre-trained** depth model **on the kitti dataset** provided by MobileStereoNet, which improves the voxel completion accuracy. In contrast, our approach does **not rely on any pre-trained depth estimators**. We employ a **implicit depth learning** based on the LSS paradigm.*
> 2. ***Stronger mIoU and Dynamic Object Handling:***
>    *Our method achieves **higher semantic performance**, with **20.81% mIoU**—a **+0.76% gain over CGFormer**. This improvement is particularly evident for **dynamic object classes** (car +2.63%, truck +7.88%, other-vehicle +8.07%, person +2.99% etc.), where we observe a **+3.5% mIoU gain** (Table 11). These dynamic categories are critical in autonomous driving applications and reflect our method’s strength in motion-aware scene understanding.*
> 3. ***Higher Efficiency:***
>    *Our model is also significantly more efficient, with **only 52.4M parameters**, compared to **122.42M in CGFormer**. This efficiency allows for better scalability and deployment potential in real-time systems.*
>
> -----
>
> > **[Weakness 3] It would be better to unify the notation of occlusion masks in Figure 3 and Figure 4 for better clarification.**
>
> **[Answer]** *We thank the reviewer for pointing this out. We will unify the notation of occlusion masks in Figures 3 and 4 of the paper.*
>
> ------
>
> > **[Weakness 4] Based on the supplementary video materials, the occlusion masks generated from optical flow appear to have significant errors. It would be beneficial to further elaborate on the role and significance of generating occlusion masks through optical flow.**
>
>
>
> **[Answer]** *We thank the reviewer for this observation. We acknowledge that occlusion masks derived from optical flow may not be perfect, especially in complex or blurred regions, as seen in some frames of the supplementary video. In fact, the white part of the occlusion mask represents the occluded area, which is only a small part. The inaccurate optical flow enhances the robustness of the model to a certain extent.*
>
> *We have designed our system to be **robust to moderate flow errors** by incorporating:*
>
> 1. ***Cosine-Similarity Weights Selection**, in 2D features, we use the cosine similarity between the current frame feature and the warped historical frame feature to design the weight. When the warp historical frame feature $F_{warp}$ is close to the current feature $F_t$, a larger weight is assigned, otherwise a smaller weight (i.e. when **the optical flow error is large**) is assigned **(see Sec. 3.4).***
>
> 2.  ***Consistency Check,** we use the forward and backward consistency check of the optical flow  to query unreliable areas in 3D space **(see Sec. 3.3 and 3.5).***
> 3. ***Empirical Effectiveness,** The occlusion cross-attention in FGTA improves mIoU by **+0.9%** (Table 5, Variant 8 vs. 4). The OGVR module further enhances performance by **+0.5%** mIoU (Table 5, Variant 8 vs. 7).*
>
> *Our model integrates historical features of **non-occluded (high confidence)** areas to enhance robustness. We have verified the effectiveness of accurate optical flow for SSC tasks. In the future, we will consider **joint optimization** of optical flow and SSC to alleviate the limitations of optical flow pre-training models. In addition, we will further explain the role and significance of generating occlusion masks through optical flow in the paper.*
>
>
>
> ------
>
> > **[Question 1] It would be better to provide more evidence to address my concerns about the role of occlusion masks in refining voxel features and the limited improvement on the KITTI360 test set. I would raise the evaluation score when these concerns are solved.**
>
> **[Answer]** *Thank you for the reviewer's thoughtful questions. We have answered your questions above and hope to have addressed them. If you have any further questions, please let us know so that we can address them as soon as possible.*
>
> ------
>
> *We thank the reviewer again and hope our clarifications address the concerns.*

---

### Official Review · Reviewer_RB8s · 2025-07-01

**Clarity:** 3
**Significance:** 2
**Originality:** 2
**Rating:** 4
**Confidence:** 3

**Summary:**

This paper introduces FlowScene, a novel method for temporal 3D Semantic Scene Completion (SSC) that leverages optical flow to improve performance. The authors argue that previous temporal SSC methods, which often just stack features from multiple frames, fail to capture crucial motion dynamics and temporal consistency. To address this, FlowScene incorporates two key modules: a Flow-Guided Temporal Aggregation (FGTA) module that uses optical flow to warp and intelligently aggregate features from past frames, and an Occlusion-Guided Voxel Refinement (OGVR) module that uses occlusion masks (derived from flow consistency checks) to refine the 3D voxel representation. This approach explicitly models motion and handles occlusions, leading to more accurate geometric and semantic predictions. The method achieves state-of-the-art results on the SemanticKITTI and SSCBench-KITTI-360 benchmarks, demonstrating significant improvements, especially in the completion of dynamic objects.

**Questions:**

1. In the Flow-Guided Temporal Aggregation (FGTA) module, you use cosine similarity between the warped historical features (F _warp) and the current features (F_t) to generate weights for aggregation. Could you provide more intuition on why this is effective? Does this implicitly down-weight regions where the flow is inaccurate (leading to poor warping and low similarity) or regions that have changed in appearance for reasons other than motion?

2. In Table 6, you compare your "Flow-Guided" alignment against a "Stack" baseline. How exactly is this "Stack" baseline implemented? Does it simply concatenate the features from the current and two historical frames along the channel dimension before the view transformation?

**Ethical Concerns:**

["NO or VERY MINOR ethics concerns only"]

**Final Justification:**

Thanks the authors for the responses. It well resolves my problems. I will keep my rating.

**Limitations:**

Yes

**Quality:**

2

**Strengths And Weaknesses:**

Strengths

1. The core concept of using optical flow to guide temporal feature aggregation and explicitly handle occlusions in 3D SSC is a strong and intuitive contribution. It directly addresses a clear limitation of prior work—the naive handling of temporal information and scene dynamics.

2.  Despite the added complexity of optical flow estimation, the proposed method is shown to be efficient. It achieves its state-of-the-art results with fewer parameters than many competing methods and maintains a comparable inference time. The use of only two historical frames also makes it practical.

3. The paper is well-written and easy to follow. The figures (especially Figures 2 and 3) effectively illustrate the core differences between FlowScene and previous methods, and the qualitative results in Figures 6 and 10 clearly showcase the method's superiority in handling occlusions and dynamic objects.

Weakness

1. The method's performance is inherently tied to the quality of the pre-trained optical flow network. The authors acknowledge this as a limitation, noting that performance could be limited in visually degraded conditions. While the ablation study shows robustness to different flow networks, a failure of the flow estimator would likely cause a catastrophic failure in the SSC prediction.

2. The paper does not report error bars or standard deviations for its results, citing computational expense. This makes it difficult to assess the statistical significance of the reported improvements over other methods.

---

> ### Author Rebuttal · Authors · 2025-07-28
>
> *We thank the reviewer for the thoughtful comments and valuable feedback. We address each concern below.*
>
>
>
> ------
>
> >  **[Weakness 1] The method's performance is inherently tied to the quality of the pre-trained optical flow network. The authors acknowledge this as a limitation, noting that performance could be limited in visually degraded conditions. While the ablation study shows robustness to different flow networks, a failure of the flow estimator would likely cause a catastrophic failure in the SSC prediction.**
>
> **[Answer]** *We acknowledge that the performance of our method is influenced by the accuracy of the optical flow estimator. However, we have designed our system to be **robust to moderate flow errors** by incorporating:*
>
> 1. ***Cosine-Similarity Weights Selection**, in 2D features, we use the cosine similarity between the current frame feature and the warped historical frame feature to design the weight. When the warp historical frame feature $F_{warp}$ is close to the current feature $F_t$, a larger weight is assigned, otherwise a smaller weight (i.e. when **the optical flow error is large**) is assigned **(see Sec. 3.4).***
>
> 2.  ***Consistency Check,** we use the forward and backward consistency check of the optical flow  to query unreliable areas and assign lower weights in 3D space **(see Sec. 3.3 and 3.5).***
>
> *Furthermore, our **flow network (GMFlow)** is lightweight yet strong **(Table 8)** . In future work, we will try to jointly optimize the optical flow and SSC tasks to alleviate the reliance on pre-trained optical flow models.*
>
>
>
> ------
>
> >  **[Weakness 2] The paper does not report error bars or standard deviations for its results, citing computational expense. This makes it difficult to assess the statistical significance of the reported improvements over other methods.**
>
> **[Answer]** *We appreciate the reviewer’s concern. We followed prior works such as MonoScene, TPVFormer, BRGScene, CGFormer and HTCL in reporting **deterministic evaluations** on fixed validation/test splits.*
>
> *Thank you for raising this important point. Following your suggestion, we conducted a statistical analysis on the SemanticKITTI validation set and report the **weighted mean IoU (W-mIoU), weighted standard deviation (W-SD), and weighted standard error (W-SE)** across semantic categories.*
>
> | Method                | mIoU(⬆️)   | W-mIoU(⬆️) | W-SD(⬇️)   | W-SE(⬇️)  |
> | :-------------------- | :-------- | :-------- | :-------- | :------- |
> | **Ours**              | **17.70** | **33.11** | **13.80** | **6.50** |
> | CGFormer (NIPS'2024)  | 16.63     | 31.91     | 14.43     | 6.80     |
> | HTCL (ECCV'2024)      | 17.08     | 32.64     | 14.17     | 6.67     |
> | BRGScene (IJCAI'2024) | 15.35     | 30.22     | 14.01     | 6.60     |
> | TPVFormer (ICCV'2023) | 12.32     | 24.44     | 14.34     | 6.75     |
> | MonoScene (CVPR'2022) | 11.08     | 22.58     | 14.38     | 6.77     |
>
> *As shown, our method not only achieves the **highest mIoU and W-mIoU**, but also demonstrates **W-SD** and **W-SE** compared to other strong baselines. This indicates that our performance improvements are **statistically meaningful and stable across classes**.*
>
> *We appreciate your suggestion to quantify this, and will include this analysis in the paper to better support our claims.*
>
>
>
> ------
>
> >  **[Question 1] In the Flow-Guided Temporal Aggregation (FGTA) module, you use cosine similarity between the warped historical features ($F _{warp}$) and the current features ($F_t$) to generate weights for aggregation. Could you provide more intuition on why this is effective? Does this implicitly down-weight regions where the flow is inaccurate (leading to poor warping and low similarity) or regions that have changed in appearance for reasons other than motion?**
>
> **[Answer]** *Yes, the use of **cosine similarity** between the warped historical features ($F_{warp}$) and the current features ($F_t$) acts as an implicit confidence estimator for flow-guided alignment. When the flow is accurate, $F_{warp}$≈$F_t$ and their similarity is high, resulting in **larger weights** for reliable regions. In contrast, misaligned or occluded areas typically produce low similarity, leading to **automatic down-weighting** during aggregation. This not only filters out flow errors but also adapts to **appearance changes** due to motion blur, lighting, or deformation. Our ablation (Table 5) confirms that this improves mIoU by **1.05%** over equal-weight aggregation (Variant 8 vs. 5).*
>
> ------
>
> >  **[Question 2] In Table 6, you compare your "Flow-Guided" alignment against a "Stack" baseline. How exactly is this "Stack" baseline implemented? Does it simply concatenate the features from the current and two historical frames along the channel dimension before the view transformation?**
>
> The **"Stack"** baseline is implemented by **concatenating the image features** from the current frame and two historical frames **along the channel dimension** prior to the view transformation step. This approach does not include any alignment or motion modeling—it treats multi-frame features as spatially aligned, which introduces artifacts when objects move. This baseline represents a **naïve temporal fusion** method and serves as a lower bound in our comparison. In contrast, our flow-guided alignment explicitly handles motion, achieving a **+2.24%** mIoU improvement over Stack (Table 6).
>
> ------
>
> *We thank the reviewer again and hope our clarifications address the concerns.*

---

> > ### Comment · Reviewer_RB8s · 2025-08-01
> >
> > Thanks the authors for the responses. It well resolves my problems. I will keep my rating.

---

> > > ### Author Response · Authors · 2025-08-01
> > >
> > > We are pleased to be able to address your concerns and thank you for your professional review.

---

### Official Review · Reviewer_r9BF · 2025-07-02

**Clarity:** 4
**Significance:** 3
**Originality:** 3
**Rating:** 5
**Confidence:** 4

**Summary:**

In this work, the authors propose a 3D Semantic Scene Completion (SSC) method tailored for autonomous driving perception tasks. Unlike existing approaches that fuse temporal information through simple stacking or attention mechanisms, the proposed method leverages estimated optical flow to align features across frames more effectively. Specifically, the authors introduce Flow-Guided Temporal Aggregation (FGTA) to aggregate historical features in the 2D space, and Occlusion-Guided Voxel Refinement (OGVR) to refine features in the 3D voxel space. Experimental results on the SemanticKITTI and SSCBench-KITTI-360 benchmarks demonstrate that the proposed method achieves state-of-the-art performance.

**Questions:**

I have a few questions about the details:

1. In the OGVR, it seems that the 2D features are projected to 3D space through the depth bins. But I do not quite get the definition or acquirement of the depth bins. Are they acquired through other depth estimation models? Or just frustum?

2. During the updating of $F_t$ in Eq.5, only the non-occluded information in historical frames is introduced for fusion. Won't the other regions help the completion? Besides, I'd also appreciate it if the authors can provide more details about the flow consistency check used in this work.

**Ethical Concerns:**

["NO or VERY MINOR ethics concerns only"]

**Final Justification:**

The rebuttal has addressed my concerns. I also hold my opinion that this work introduces a reasonable and valuable framework. Therefore, I will keep my rating.

**Limitations:**

Yes.

**Quality:**

3

**Strengths And Weaknesses:**

Strengths:

1. The motivation for introducing optical flow to guide temporal feature fusion is well-justified and conceptually sound.

2. The writing is clear, and the diagrams are well-designed and easy to follow, which helps in understanding the methodology.

3. Experimental results on benchmark datasets convincingly demonstrate the proposed method's superiority over existing approaches.

Weaknesses:

1. While the integration of optical flow into the framework is reasonable, it inherently depends on the quality of flow estimation from pre-trained networks. As shown in Table 7, aggregating more historical frames will instead degrade the performance, likely due to inaccuracies in flow estimation. This outcome contradicts intuitive expectations. It may be beneficial to explore joint optimization of the flow estimation and the SSC framework to mitigate this limitation.

2. The additional computational cost introduced by flow estimation may result in slower inference compared to other methods. However, this trade-off might be acceptable depending on the application context.

---

> ### Author Rebuttal · Authors · 2025-07-27
>
> *We sincerely thank the reviewer for the valuable feedback and insightful suggestions. Below, we address the identified weaknesses and questions in detail.*
>
>
>
> ------
>
> > **[Weakness 1] While the integration of optical flow into the framework is reasonable, it inherently depends on the quality of flow estimation from pre-trained networks. As shown in Table 7, aggregating more historical frames will instead degrade the performance, likely due to inaccuracies in flow estimation. This outcome contradicts intuitive expectations. It may be beneficial to explore joint optimization of the flow estimation and the SSC framework to mitigate this limitation.**
>
> **[Answer]** *We agree that the performance gain of our approach depends to some extent on the quality of the flow estimates. As observed in Table 7, incorporating more than two historical frames can introduce temporal drift and noise due to increased difficulty in long-range flow estimation. To address this, we empirically determine that using **two historical frames** strikes the best balance between **inference times and flow accuracy**. Furthermore, our **flow network (GMFlow)** is lightweight yet strong **(Table 8)** .*
>
> *Besides, we have designed our system to be **robust to moderate flow errors** by incorporating:*
>
> 1. ***Cosine-Similarity Weights Selection**, in 2D features, we use the cosine similarity between the current frame feature $F_t$ and the warped historical frame feature $F_{warp}$ to design the weight. When the warp historical frame feature $F_{warp}$ is close to the current feature $F_t$, a larger weight is assigned, otherwise a smaller weight (i.e. when **the optical flow error is large**) is assigned **(see Sec. 3.4).***
>
> 2.  ***Consistency Check,** we use the forward and backward consistency check of the optical flow  to query unreliable areas and assign lower weights in 3D space **(see Sec. 3.3 and 3.5).***
>
>
>
> ------
>
> > **[Weakness 2] The additional computational cost introduced by flow estimation may result in slower inference compared to other methods. However, this trade-off might be acceptable depending on the application context.**
>
> **[Answer]** *While flow estimation introduces overhead, our framework remains efficient. As shown in Table 4, our method achieves a competitive inference time of **0.301s**, outperforming or matching many temporal methods such as OccFormer (0.338s), HTCL (0.297s), and Symphonize (0.319s). This is due to our use of **lightweight GMFlow** and the **RepViT encoder**, which also reduce model size (see Tables 8, 9 and 10). We believe this trade-off is acceptable, especially given the significant performance gains in dynamic object modeling (Table 11).*
>
>
>
> ------
>
> > **[Question 1] In the OGVR, it seems that the 2D features are projected to 3D space through the depth bins. But I do not quite get the definition or acquirement of the depth bins. Are they acquired through other depth estimation models? Or just frustum?**
>
> **[Answer]** *The **depth bins** used in the OGVR module follow the **Lift-Splat-Shoot (LSS, ECCV'2020)** paradigm. We do not rely on an explicit depth estimation model. Instead, we use **discretized frustum-based depth bins**, where the depth distribution is implicitly learned using a soft depth volume and supervised via LIDAR projected depths.*
>
>
>
> ------
>
> > **[Question 2] During the updating of $F_t$ in Eq.5, only the non-occluded information in historical frames is introduced for fusion. Won't the other regions help the completion? Besides, I'd also appreciate it if the authors can provide more details about the flow consistency check used in this work.**
>
> **[Answer]**
>
> 1. *We selectively use **non-occluded features** from historical frames in Eq. (5) to prevent injecting unreliable or distorted information caused by occlusions or inaccurate flow. Including occluded regions can introduce noise and harm completion quality. Our ablation (Table 5, Variant 5 vs. 4) confirms that this selective strategy improves mIoU.*
> 2. *For **occlusion detection**, we use a **forward-backward flow consistency check** (Section 3.3, Figure 4). A pixel is marked as occluded if the discrepancy between its forward and backward flow exceeds a threshold. The resulting mask is used both in the FGTA module and for guiding voxel refinement in 3D space.*
>
> *Besides, we will add details about the consistency check in the Technical Appendix of the paper.*
>
> **Following are the details about flow consistency check:**
>
> ### 1. Variable Definition:
>
> **Forward Flow**: \( $Flow^{t \rightarrow t-1}$ \)
>
> - Maps pixels from the current frame \( $I_t $\) to the previous frame \( $I_{t-1}$ \).
>
> - For a pixel \(${x}_t \in I_t $\), the corresponding location in \($I\_{t-1}$\)  is:
>
>   $${x}_{t-1} = {x}_t + Flow^{t \rightarrow t-1}({x}_t)$$
>
> **Backward Flow**: \($ Flow^{t-1 \rightarrow t}$ \)
>
> - Maps pixels from the previous frame \($ I_{t-1} $\) to the current frame \( $I_t$ \).
>
> - For a pixel \( ${x}_{t-1} \in I\_{t-1} $\), the corresponding location in \( $I_t$ \) is:
>
>   $${x}_t' = {x}\_{t-1} + Flow^{t-1 \rightarrow t}({x}\_{t-1})$$
>
> ### 2. Consistency Check
>
> The **forward-backward consistency check** verifies whether a pixel mapping is valid by ensuring round-trip correspondence.
>
> **Round-trip Mapping**
>
> A pixel in \( $I_t $\) is mapped to \( $I_{t-1} $\) using forward flow, and then mapped back using backward flow:
>
> $${x}_t'' = {x}_t + Flow^{t \rightarrow t-1}({x}_t) + Flow^{t-1 \rightarrow t}({x}_t + F^{t \rightarrow t-1}({x}_t))$$
>
> Define the **consistency residual** as:
> $$\Delta({x}_t) = Flow^{t \rightarrow t-1}({x}_t) + Flow^{t-1 \rightarrow t}({x}_t + F^{t \rightarrow t-1}({x}_t))$$
>
> If the norm \( $\|\Delta({x}_t)\| $\) is small (below a threshold), the mapping is considered consistent.
>
> ### 3. Occlusion Mask
>
> Pixels with high inconsistency are typically considered **occluded or unreliable**.
>
> **Occlusion Mask \($ M({x}) $\):**
> $$
> M({x}) =
> \begin{cases}
> 1 & \text{if } \|\Delta({x})\| > \tau \quad \text{(occluded)} \\\\
> 0 & \text{otherwise} \quad \text{(non-occluded)}
> \end{cases}
> $$
>
> Where \($ \tau $\) is a predefined threshold.
>
>
> ------
>
> *We thank the reviewer again and hope our clarifications address the concerns.*

---

> > ### Comment · Reviewer_r9BF · 2025-08-01
> >
> > Thanks the authors for the responses. It well resolves my problems. I will keep my rating.

---

> > > ### Author Response · Authors · 2025-08-01
> > >
> > > We are pleased to be able to address your concerns and thank you for your professional review.

---

### Decision · Program_Chairs · 2025-09-17

**Decision:**

Accept (poster)

**Comment:**

The paper proposes a 3D semantic scene completion method using optical flow and occlusion cues.
The reviewers give **one accept** and **two borderline accepts**.

**Strengths**
- (r9BF,RB8s,n69C) The motivation for using optical flow for temporal feature fusion is well justified.
- (r9BF,RB8s) Good clarity
- (r9BF,RB8s,n69C) Better accuracy over existing approaches

**Weaknesses**
- (r9BF,RB8s) Accuracy depending on the off-the-shelf optical flow methods
   - (resolved) Table 8 reports the accuracy analyses on using different optical flow methods. The difference is not so substantial.
- (n69C) The unclear role of occlusion mask
   - (resolved) Justified in the rebuttal (through empirical finding)
- (n69C) Justification of limited accuracy on KITTI360
   - (resolved) Other methods have a more advantageous setup; the proposed method is more light-weighted.


The rebuttal resolves almost all of the concerns from the reviewers. ACs agree to accept the paper. It's highly encouraged to include all the new analyses and discussions in the final version.